# Numerical Analysis and Dynamic Response of Optimized Composite Cross Elliptical Pressure Hull Subject to Non-Contact Underwater Blast Loading

**Mahmoud Helal** [1,2,3,4], **Huinan Huang** [1,2], **Elsayed Fathallah** [1,2,5,*] , **Defu Wang** [1,2,*], **Mohamed Mokbel ElShafey** [6] **and Mohamed A. E. M. Ali** [5]

[1]  College of Engineering, Northeast Agricultural University, Harbin 150030, Heilongjiang, China
[2]  Key Laboratory of Swine Facilities Engineering, Ministry of Agriculture, Harbin 150030, Heilongjiang, China
[3]  Production and Mechanical Design Dept., Faculty of Engineering, Mansoura University, Mansoura 35516, Egypt
[4]  Department of Mechanical Engineering, Faculty of Engineering, Taif University, Taif 21974, Saudi Arabia
[5]  Department of Civil Engineering, Military Technical College, Cairo 11865, Egypt
[6]  Civil Engineering Department Canadian International College, Cairo 11865, Egypt
*   Correspondence: saidhabib2000@hotmail.com (E.F.); dfwang0203@163.com (D.W.);
    Tel.: +20-1-09900177 or +20-1-12341914 (E.F.)

**Abstract:** Among the most important problems confronted by designers of submarines is to minimize the weight, increase the payload, and enhance the strength of pressure hull in order to sustain the hydrostatic pressure and underwater explosions (UNDEX). In this study, a Multiple Intersecting Cross Elliptical Pressure Hull (MICEPH) subjected to hydrostatic pressure was first optimized to increase the payload according to the design requirements. Thereafter, according to the optimum design results, a numerical analysis for the fluid structure interaction (FSI) phenomena and UNDEX were implemented using nonlinear finite element code ABAQUS/Explicit. The propagation of shock waves through the MICEPH was analyzed and the response modes (breathing, accordion and whipping) were discussed. Furthermore, the acceleration, displacement and failure index time histories at different locations were presented. The results showed that the greatest acceleration occurred in the athwart direction, followed by the vertical and longitudinal directions. Additionally, the first bubble pulse has a major effect on athwart acceleration. Moreover, the analysis can be effectively used to predict and calculate the failure indices of pressure hull. Additionally, it provides an efficient method that reasonably captures the dynamic response of a pressure hull subjected to UNDEX.

**Keywords:** underwater explosion; composite pressure hull; whipping; breathing; failure index

## 1. Introduction

Significant research work has been presented in order to simulate the behavior of underwater vehicles under severe loading conditions. For instance, Reddy [1] illustrated the effect of shock pressure loading on a ring-stiffened submersible hull using finite element analysis. The failure analysis indicated that fibers failed in tension while matrix failed in shear when the explosion charge exceeded 25 kg TNT. Furthermore, Jen [2] worked on minimizing the weight of pressure hull by enhancing the pressure hull strength taking into consideration both hydrostatical pressure and underwater explosion. Additionally, the results indicated that the dynamic motion of the pressure hull has an accordion mode, a whipping mode and a breathing mode. Also, Cho et al. [3] derived an empirical formula for predicting the collapse strength of composite cylindrical structures under hydrostatic pressure as a function of important design parameters such as the geometric dimensions and the layered angle.

In addition, Chen et al. [4] experimentally investigated the dynamic performance of ship body coated by rubber. The results demonstrated that the coating was not effective at reducing the low-frequency whipping motion excited by the bubble pulse. However, it was able to moderate the high-frequency response excited by shock wave. Moreover, Ramajeyathilagam et al. [5] numerically and experimentally investigated the effect of underwater explosions (UNDEX) on rectangular plates. It was revealed that the underwater explosion failures can be predicted using the strain rate effects. Also, Liu et al. [6] illustrated the global responses of ship subjected to UNDEX. The results demonstrated that the UNDEX waves changed the added masses on the ship and effectively affected the global responses of its body. On the other hand, Liang et al. [7] examined the transient dynamic responses of submarine pressure hull exposed to hydrostatic pressure and shock loading. It was concluded that the collapse depth (maximum diving) of the submarine pressure hull was about 700 m. In addition, they observed that the loading condition depends not only on the hydrostatic pressure but also on the shock loading. Likewise, Kwon and Fox [8] studied experimentally and numerically the dynamic response of a cylinder subjected to UNDEX. The results illustrated that the largest strains on a cylinder subjected to a far-field side-on UNDEX occurred near the two ends on the near side of the cylinder to a far field explosive charge. Additionally, the damage occurred at the center on the opposite side of the cylinder. Similarly, McCoy and Sun [9] combined FSI code and finite element modeling techniques to investigate the dynamic response of a thick-section hollow composite cylinder. The results showed that the fluid-structure coupling has a significant effect on stress distributions within the structure.

Furthermore, Shin and Hooker [10] predicted numerically the damage response of submerged imperfect cylindrical structures exposed to UNDEX. Based on these results, the introduction of initial imperfections greatly affected the response of the cylinder when compared with the response of a perfect cylinder. On the other hand, Qiankun and Gangyi [11] demonstrated the shock response of a ship section to non-contact UNDEX using the finite element software package ABAQUS. It was revealed that the fluid thickness and size of fluid mesh effectively affects and improve the modeling accuracy. Additionally, Adamczyk and Cichocki [12,13] performed a numerical study to obtain the shock response of an underwater hybrid structure subjected to UNDEX. Zhao et al. [14] predicted the damage features of RC slabs subjected to air and UNDEX. The shock wave propagation and damage mechanisms from contact explosions in air and water were compared. The dynamic response of the RC slab is highly localized in the air contact explosion. Furthermore, the crater failure is observed at the top surface of the RC slab due to the direct impact of the air contact blast loading and the spalling failure occurs at the bottom of the RC slab. On the other hand, when the RC slab subjected to underwater contact explosion the top surface of the reinforced concrete RC slab almost completely destroyed. Therefore, underwater contact explosion can cause significantly more damage to the RC slab than the same amount of explosive in air. Rajendran and Narasimhan [15] investigate the damage of clamped circular plates subjected to contact UNDEX. The deformation contours were a spherical viewing the maximum absorption of energy for the depth of bulge attained. Also, Jacinto et al. [16] applied a linear dynamic analysis of plate models under explosions. The element size has a great effect on the results. Likewise, Kumar et al. [17] experimentally studied the blast effect on carbon composite panels the results showed that. There were two types of dominant failure mechanisms observed, fiber breakage and inter-layer delamination. Furthermore, Guo et al. [18] presented a new shock factor of twin hull water plane (catamaran) subjected to UNDEX. The shock factor parameter is used to describe the response of ships exposed to this loading condition based on shock wave energy. Similarly, Wang et al. [19] proposed a new method involving an analytical technique connected to elastic dynamic response of laminated plates exposed to UNDEX. The method was validated by comparing its results with those achieved by a semi analytical method and the experiment results. Zhang et al. [20] predicted the dynamic bending moment of a UNDEX bubble acting on a hull. The predicted numerical results showed that UNDEX bubble propagated a longitudinal bending which caused sagging and hogging damage for the ship. Furthermore, Gong and Khoo [21] presented a transient response of an UNDEX bubble on a glass/epoxy composite deep-submersible pressure hull. It was observed that the UNDEX

bubble produced a huge deformation around the stand-off point in the pressure hull immediately after the collapse of the bubble, and the minimum volume was observed beneath the composite hull. On the other hand, Wang et al. [22] investigated numerically and experimentally the failure mode and dynamic response of a ship structure subjected to shock wave and bubble pulse. Jun et al. [23] investigated the impact environment characteristics of floating shock platform subject to UNDEX. Young and Leonard [24] modeled and simulate a surface ship shock to UNDEX, the results showed that the cavitation effect must be taken into account in the ship shock simulation, and that cavitation volume must be large enough. Gannon [25] investigated the response of a submerged stiffened cylinder to UNDEX using a coupled Eulerian Lagrangian model and experimental approaches.

This present study developing a procedure and describes a numerical modelling methodology for calculating the dynamic response of optimized Multiple Intersecting Cross Elliptical Pressure Hull (MICEPH) exposed to non-contact UNDEX. First, a submarine with pressure hull in the form of MICEPH is optimized using non-linear finite element analysis software ANSYS. Thereafter, according to optimization, the finite element model is built using non-linear finite element code ABAQUS/Explicit to examine the dynamic response of the pressure hull exposed to non-contact UNDEX.

## 2. Analysis of Underwater Shock Loading and Bubble Pulse

The most important element of any submersible body is the pressure hull. It contributes about one-fourth to one-half of the total underwater vehicle weight. Figure 1 presents various pressure hull configurations used in submersible bodies [26–28]. Exposing the hull structure to shock wave and bubble pulsation upon UNDEX events leads to great damage to the hull structure [2]. Figure 2 presents the different configurations that occur during UNDEX events [29,30]. The loading mechanisms resulting from UNDEX include incident wave, free surface reflection, shockwave, bottom reflection wave, gas bubble oscillation, bubble-pulse loading and bulk cavitation [31]. The compressive load on the structure of the hull is increased due the reflections from the bottom of the ocean which propagates from the shock wave, while the reflection of the shock wave from the ocean's free surface causes a reduction in the pressure [32,33]. Equation (1) presents the evaluation technique of pressure time history ($P_{in}(t)$) for the pressure profile, as follows [2,34–38]:

$$P_{in}(t) = P_{max}e^{-\left(\frac{t-t_1}{\theta}\right)} \quad (\text{Mpa})? \quad t \geq t_1 \tag{1}$$

where ($t$) denotes the time elapsed after detonation of charge (ms), ($P_{max}$) denotes the peak pressure (Mpa), ($t_1$) denotes the arrival time of shock wave to the target after the detonation of the charge (ms), and ($\theta$) denotes the time decay constant that describes the exponential decay (ms). The peak pressure and decay constant depends on the size of the explosion and the stand-off distance from the charge at which the pressure is measured. The peak pressure ($P_{max}$), decay constant of the wave ($\theta$), impulse ($I$), bubble oscillation period ($T$), the maximum radius of the first bubble of explosive gas ($R_{max}$), and the energy flux density/energy per unit volume ($E$) can be expressed as per Equations (2)–(7) [39–41].

$$P_{max} = K_1\left(\frac{W^{\frac{1}{3}}}{R}\right)^{A_1} \quad (\text{Mpa}) \tag{2}$$

$$\theta = K_2W^{1/3}\left(\frac{W^{\frac{1}{3}}}{R}\right)^{A_2} \quad (\text{ms}) \tag{3}$$

$$I = K_3W^{1/3}\left(\frac{W^{\frac{1}{3}}}{R}\right)^{A_3} \quad (\text{Mpa-sec}) \tag{4}$$

$$T = K_5 \frac{W^{\frac{1}{3}}}{(D + 9.8)^{5/6}} \quad (\text{sec}) \tag{5}$$

$$R_{max} = K_6 \left( \frac{W}{(D + 9.8)} \right)^{1/3} \quad (\text{m}) \tag{6}$$

$$E = K_4 W^{\frac{1}{3}} \left( \frac{W^{\frac{1}{3}}}{(R)} \right)^{A_4} \quad (\text{m} \cdot \text{kPa}) \tag{7}$$

where $(K_1, K_2, K_3, K_4, K_5, K_6)$ and $(A_1, A_2, A_3, A_4)$ are constants that depend on the explosive charge type with values as in [39,42]. These input constants are illustrated in Figure 2. Phenomena of the underwater explosions (UNDEX): shock wave, high pressure and bubble motion.

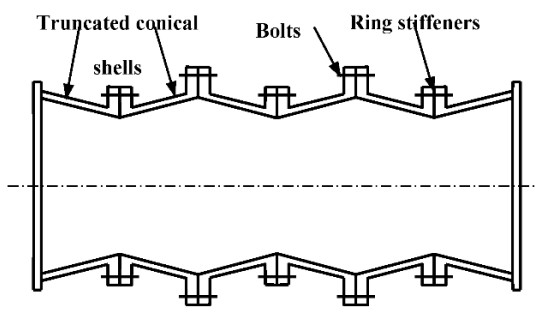

(**a**) Ring-stiffened corrugated pressure hull.

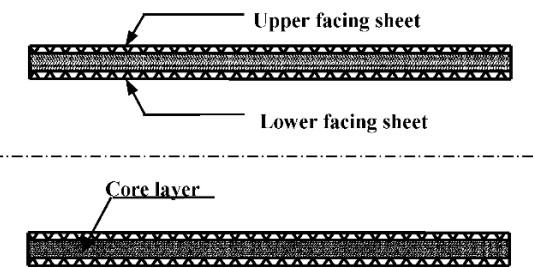

(**b**) Sandwich hull wall construction.

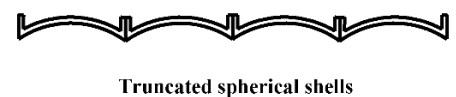

(**c**) Modular composite pressure hull made of truncated spherical section.

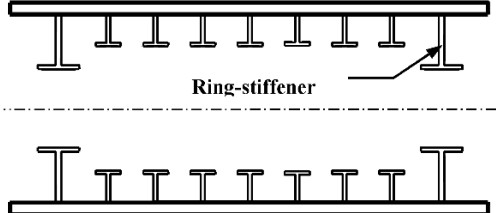

(**d**) Gas-storage toroidal tube-stiffener hull wall construction.

(**e**) Traditional ring-stiffened hull wall construction.

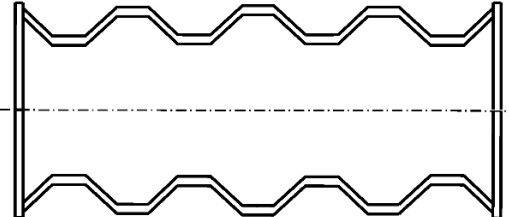

(**f**) Trapesoid corrugated hull wall construction.

**Figure 1.** Various wall structures utilized for pressure hulls. (**a**) Reproduced with permission from [26], Copyright Elsevier, 2011. (**b,d–f**) reproduced with permission from [27], Copyright Elsevier, 2003. (**c**) reproduced with permission from [28], Copyright Elsevier, 2016.

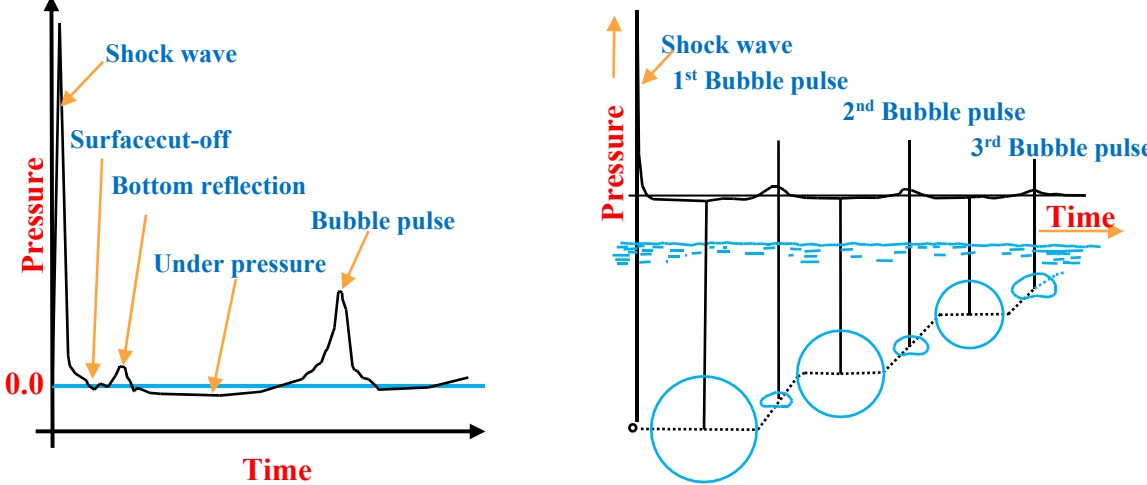

**Figure 2.** Phenomena of the underwater explosions (UNDEX): shock wave, high pressure and bubble motion.

(*W*) denotes the mass of the explosive charge in (Kg), (*D*) denotes the charge depth in *m*, and (*R*) denotes the distance between the explosive charge and target in (m). In UNDEX events, there are two types of subsequent cavitations that occur: the bulk and the local cavitation. The latter is caused by the reflection of the shock wave at the free surface and must be taken into consideration during the analysis of the surface ship. When the shock wave impinges upon the structure, then the total pressure $P_{tot}(t)$ at the fluid structure interface can be expressed according to Equation (8) [43]:

$$P_{tot}(t) = P_{in}(t) + P_c(t) + P_{st} \tag{8}$$

where $P_{in}(t)$ denotes the incident shock wave, $P_c(t)$ denotes the scattered pressure, and $P_{st}$ denotes the hydrostatic pressure. Accordingly, the local cavitation occurs in the water as the pressure drops to vapor pressure (about 0.3 psi) [31]. Thereafter, the cavitation collapses and reload the structure.

## 3. Composite Failure Criteria

An appropriate failure criterion is needed in order to find the maximum permissible load before lamina failure. Therefore, it is necessary to develop theories to compare the state of stresses and strains in materials [44–46].

### 3.1. Maximum Stress Failure Criteria

The maximum stress criteria are single mode failure criteria. Fracture occurs if stresses at the principal material coordinates are higher than their respective strength. The failure index ($I_F$) is presented in Equation (9) [47]:

$$I_F = \max \begin{cases} \sigma_{11}/X_t & \text{if } \sigma_{11} > 0 \text{ } or -\sigma_{11}/X_c \text{ if } \sigma_{11} < 0 \\ \sigma_{22}/Y_t & \text{if } \sigma_{22} > 0 \text{ } or -\sigma_{22}/Y_c \text{ if } \sigma_{22} < 0 \\ |\tau_{12}|/S \end{cases} \tag{9}$$

where $X_t$, $X_c$, $Y_t$, $Y_c$ and $S$ denote the ultimate longitudinal, transversal and shear strength constants, respectively. Also, ($\sigma_{11}$, $\sigma_{22}$ and $\tau_{12}$) denote the applied longitudinal, transversal and shear stress components, respectively, and can be calculated using Equation (10) [48]:

$$\begin{Bmatrix} \sigma_{11} \\ \sigma_{22} \\ \tau_{12} \end{Bmatrix} = \begin{bmatrix} Q_{11} & Q_{12} & 0 \\ Q_{12} & Q_{22} & 0 \\ 0 & 0 & Q_{66} \end{bmatrix} \begin{Bmatrix} \varepsilon_{11} \\ \varepsilon_{22} \\ \gamma_{12} \end{Bmatrix} \tag{10}$$

where $Q_{11} = (1 - v_{12}v_{21})^{-1}E_1$, $Q_{12} = (1 - v_{12}v_{21})^{-1}E_1v_{21}$, $Q_{22} = (1 - v_{12}v_{21})^{-1}E_2$, $Q_{66} = G_{12}$.

### 3.2. Tsai-Hill Failure Criteria

Tsai-Hill failure criterion assume that there is an interaction between longitudinal, transversal and shear strength in the damage progress. The Tsai-Hill failure criterion can be expressed as in Equation (11) [49]:

$$\sigma_{11}^2/X^2 + \sigma_{22}^2/Y^2 - \sigma_{11}\sigma_{22}/X^2 + \tau_{12}^2/S^2 = 1 \tag{11}$$

where $\sigma_{11}$, $\sigma_{22}$ and $\tau_{12}$ are the applied longitudinal, transversal and shear stress components, respectively, and can be calculated using Equation (10). Meanwhile, $X$ and $Y$ are the longitude and traverse strength, respectively, whether in tension or compression, which depends on the stress status in the laminates. $S$ is the shear strength constant.

### 3.3. Tsai-Wu Failure Criteria

The Tsai-Wu failure criterion is the most generalized criterion that distinguishes between compressive and tensile strength. The criterion can be expressed as shown in Equation (12) [50,51]:

$$FI = \sigma_{11}\left(\frac{1}{X_t} - \frac{1}{X_c}\right) + \sigma_{22}\left(\frac{1}{Y_t} - \frac{1}{Y_c}\right) - \frac{\sigma_{11}^2}{X_t \times X_c} - \frac{\sigma_{22}^2}{Y_t \times Y_c} - \frac{\tau_{12}^2}{S^2} \tag{12}$$

where $FI$ is the failure index, $X_t$, $Y_t$, $X_c$, $Y_c$, $\sigma_{11}$, $\sigma_{22}$, $\tau_{21}$ and $S$ are as aforementioned. The failure occurs when the calculated stresses reaches the ultimate stresses and the $FI$ reaches or exceeds the value 1 [52]. In finite element procedures, failure criteria are presented using a defined failure index and can be presented as per Equation (13):

$$FI = \frac{Stress}{Strength} \tag{13}$$

The results presented in this work are based on maximum stress, Tsai-Hill and Tsai-Wu failure criteria.

## 4. Optimization and Geometrical Configuration of MICEPH

In this work, the proposed form of submarine pressure hull is a multiple intersecting cross-elliptical hull, constructed from Carbon/Epoxy composite (USN-150) with stacking sequence $[(\alpha/-\alpha)_4]_s$. The optimization is performed using the ANSYS V14.5 (ANSYS, Canonsburg, PA, USA) parametric design language (APDL) to maximize buckling load and minimize the buoyancy factor under the constraints of failure strength and deflection ($\delta_{max}$) according to the design requirements. The model was built using SHELL99 for the shell and BEAM189 for the ring and long beams [53,54]. The material properties and the strength parameters are presented in Table 1 [55]. Figure 3 shows the proposed MICEPH utilized in this study. Figure 3 shows the multi-objective optimization procedure flow chart. The random design generation method (RDGM), which is a sub type of the sub problem approximation used in (ANSYS) will be considered in this study. On the other hand, Table 2 illustrates the results of the multi-objective optimization for MICEPH which indicates the optimal design point and objective function. The table contains the failure index ($FI$) for the maximum stress ($F_{MAXF}$) and Tsai-Wu ($F_{TWSR}$) failure criteria. The optimal objective function ($MOF$) is 0.0487, the buoyancy factor ($B.F$) is 0.197, and the buckling strength factor is 4.056, with a total hull weight of 394.57 kg. The optimal angle-ply orientation ($\alpha$) is 49°, with a layer thickness ($t$) equal to 1.294 mm. The major diameter ($D_{major}$) and minor diameter ($D_{minor}$) are 2.0 m and 1.6836 m, respectively. The intersecting angel ($\theta$) is 40°, and the radius ($R$) is equal to 0.50 m.

**Table 1.** Strengths of unidirectional composites and material properties of the sandwich components.

| Material | Material and Strength Properties |
|---|---|
| **Carbon/epoxy composite (USN-150)** | $E_{11}$ = 131 GPa, $E_{22}$ = 10.8 GPa, $E_{33}$ = 10.8 GPa, $G_{12}$ = 5.65 GPa, $G_{23}$ = 5.65 GPa, $v_{12}$ = 0.28, $v_{23}$ = 0.0.59, $X_t$ = 2000 MPa, $X_c$ = 1400 Mpa, $Y_t$ = 61 MPa, $Y_c$ = 130 MPa, $S$ = 70 MPa, $\rho$ = 1540 kg/m$^3$ |

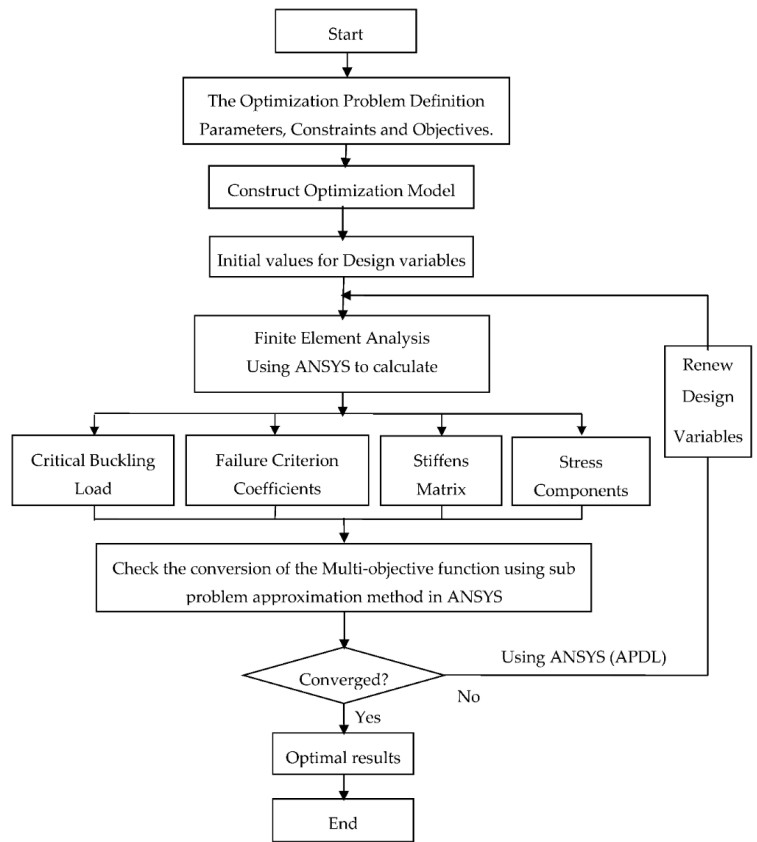

**Figure 3.** Multi-objective optimization procedure flow chart.

**Table 2.** The results of optimal design for pressure hull without core (Carbon/epoxy composite (USN)).

| | Tsai-Wu Failure ($F_{TWSR}$) | Maximum Stress Failure ($F_{MAXF}$) | | Tsai-Wu Failure ($F_{TWSR}$) | Maximum Stress Failure ($F_{MAXF}$) | Buckling Strength Factor ($\lambda$) | 4.05636769 |
|---|---|---|---|---|---|---|---|
| *FI_1* | 0.741309805 | 0.678269306 | *FI_12* | 0.821134469 | 0.752770265 | *Layer thickness (t)* | 1.2946 mm |
| *FI_2* | 0.722204168 | 0.663666634 | *FI_13* | 0.862284155 | 0.786597570 | *B.F* | 0.197617464 |
| *FI_3* | 0.719470819 | 0.648617196 | *FI_14* | 0.870768505 | 0.827885448 | **Total weight** | 394.574729 kg |
| *FI_4* | 0.701161177 | 0.639770414 | *FI_15* | 0.915660726 | 0.858129679 | $\theta$ | 40° |
| *FI_5* | 0.698515548 | 0.634590148 | *FI_16* | 0.964664573 | 0.999689193 | $\alpha$ | 49° |
| *FI_6* | 0.689078750 | 0.625456838 | $D_{major}$ | 2.0 m | | $h_1$ | 75.4 mm |
| *FI_7* | 0.705410043 | 0.620743340 | $D_{minor}$ | 1.6836 m | | $b1$ | 56 mm |
| *FI_8* | 0.707480479 | 0.671368673 | **Operating depth (H)** | 500 m | | $h_2$ | 50 mm |
| *FI_9* | 0.698181393 | 0.664485365 | **Maximum deflection ($\delta_{MAX}$)** | 4.65067861 (mm) | | $b2$ | 5 mm |
| *FI_10* | 0.772656298 | 0.678194589 | **R** | 0.50 m | | $h_3$ | 50 mm |
| *FI_11* | 0.810096248 | 0.714633754 | *MOF* | $4.871783801 \times 10^{-2}$ | | $b3$ | 5 mm |

## 5. Modeling and Simulation of MICEPH

The fluid structure interaction (FSI) and UNDEX phenomena are implemented in non-linear finite element code ABAQUS V6.13 (Dassault Sys Simulia Corp, Providence, RI, USA)/Explicit. The MICEPH used in this study consists of five cross-ellipses, as demonstrated in Figure 4. The optimized structural hull and water domain are imported from ANSYS Program as (* iges). Since the water domain does not have internal space for the pressure hull, it therefore needs to be modified to have identical internal shape for the hull. This can be simply done using the merge/cut function in the CAE model part in ABAQUS. The pressure hull was modelled using the shell elements. The stiffeners were added in the radial and longitudinal directions to provide structural integrity. The fourth node shell element (S4R) was used to model the pressure hull and stiffeners. An assemblage of 4-node acoustic tetrahedral elements (AC3D4) was used to represent the external fluid. The total horizontal length of the fluid model with the spherical ends was 11.56 m. The vertical length of the fluid domain was 7 m. The modeled water domain part includes the hull structure. The model and the charge mass were located at depth 100 m below the free surface. In dynamic problems that involve fluid and coupled solid medium, the interface between the two domains must be identical. Also, the water domain must have bulk modulus and density since it is acoustic domain. Figure 5 shows the finite element model and the meshing technique for the MICEPH surrounded by the fluid. The boundaries of the fluid around the MICEPH may cause shock wave reflection or refraction, which may cause a change in its superposition or cancellation by the incident wave [56,57]. To overcome this problem, the boundary condition of the fluid is executed as a non-reflective boundary condition during the analysis. The pressure hull was exposed to UNDEX produced by various amounts of explosives and offset distances (30 kg TNT at 5.5 m, 20 kg TNT at 5.5 m, 15.5 m and 20.5 m). In this study, the stand-off distance is located at the right side of the cross-elliptical submersible pressure hull. The stand-off point represents the location where the incident wave defined and represented by reference point1 ($RP_1$). The location of the charge (source point) defined as reference point 2 ($RP_2$), which represents the position of the charge as illustrated in Figure 6. There are three types of input parameters for UNDEX: physical charge, material and bubble model. Defining the UNDEX bubble in the interaction module was applied by defining the source point, stand off point which specifying the wave properties and charge depth. Taking into consideration that the infinite surface shouldn't reflect the shock wave; thus, acoustic impedance (non-reflecting) has to be applied. The initial boundary condition for the infinite surface of water domain (acoustic wave) was set to zero, which means that the domain is calm water, i.e., no interference [58].

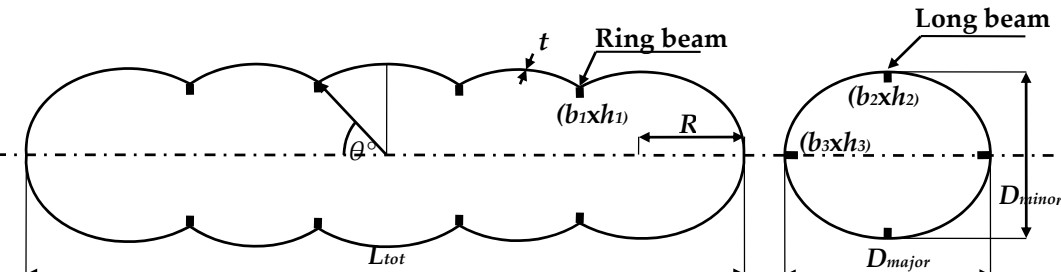

**Figure 4.** Miceph.

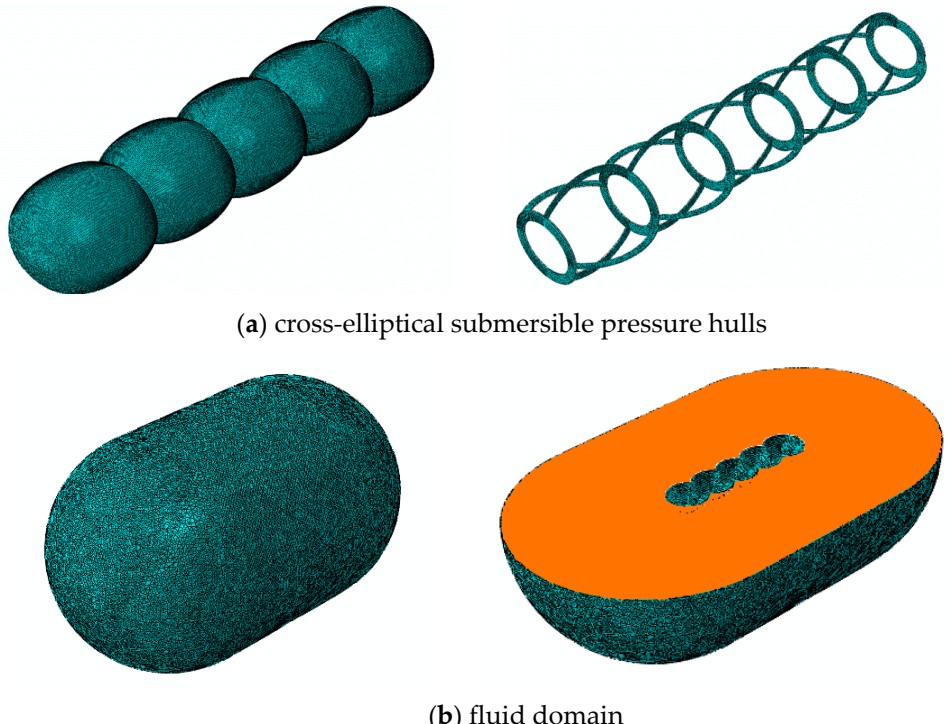

(**a**) cross-elliptical submersible pressure hulls

(**b**) fluid domain

**Figure 5.** Finite element modeling of: (**a**) cross-elliptical submersible pressure hulls, and (**b**) fluid domain.

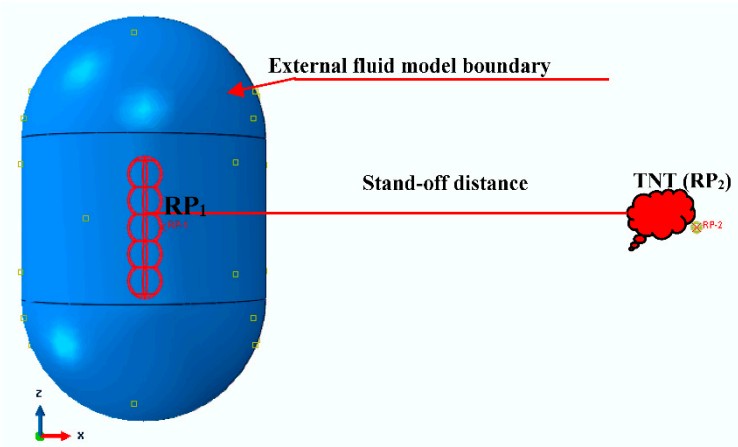

**Figure 6.** The location of charge (source point (RP$_2$)) and stand-off point (RP$_1$).

## 6. Results and Analysis

### 6.1. Propagation of Shock Wave

In this study, the MICEPH is subjected to shock wave and bubble pulse owing to UNDEX. Its performance depends mainly on the strength of the waves and the resilience of its structure. The stand-off point at node (*A*) was first struck by the shock wave. The charge is located at different stand-off distances—5.5 m, 15.5 m and 20.5 m—measured from the stand-off point. In the modelling procedures, two output variables were provided for the acoustic pressure: (i) the acoustic pressure (POR), which represents the total dynamic pressure in the wave formulation analysis including additional pressure induced by the incident and scattered waves; and (ii) the absolute acoustic pressure (PABS), which is the sum of the acoustic pressure and hydrostatic pressure. The acoustic pressure magnitudes are in Pascal [56]. Figure 7 demonstrates the POR at which the total dynamic pressure

instantaneously occurs at the time of explosion (zero time instant) due to total wave formulation. Also, the figure shows the spherical shape of the shock wave and the initial instantaneous shock wave propagation at the time of the explosion due to different explosive weights and stand-off distances. It can be observed that the maximum shock wave pressure determined was $3.471 \times 10^7$ Pa due to 30 kg TNT at offset distance of 5.5 m. Figure 7b–d illustrates the computed POR field distribution due to 20 kg TNT at offset distances of 5.5 m, 15.5 m and 20.5 m. The maximum shock wave pressure measured was $2.373 \times 10^7$ Pa, $6.777 \times 10^6$ Pa and $4.853 \times 10^6$ Pa at explosive offset distance 5.5 m, 15.5 m and 20.5 m, respectively. The location of the source and standoff point greatly affect the computed POR field distribution, propagation and its magnitude.

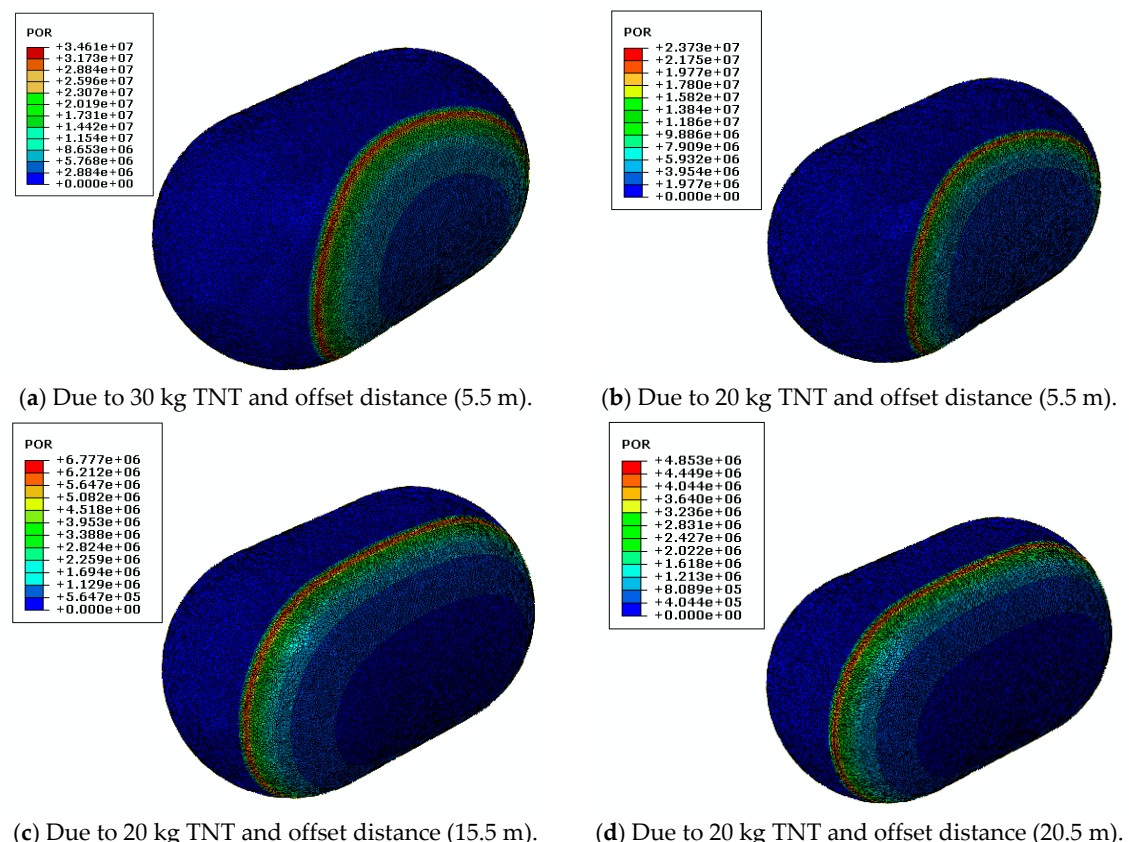

(**a**) Due to 30 kg TNT and offset distance (5.5 m).　　　(**b**) Due to 20 kg TNT and offset distance (5.5 m).

(**c**) Due to 20 kg TNT and offset distance (15.5 m).　　　(**d**) Due to 20 kg TNT and offset distance (20.5 m).

**Figure 7.** Computed POR field at the zero time instant due to total wave formulation.

Figures 8 and 9 present the computed POR field distribution at several time instants at the front and back sides of fluid domain. The propagation of the shockwave and the dynamic pressure in the surrounding water were clearly presented in these figures. The figures demonstrated that there are some radial POR waves that hits the structure and consequently leads to some deformations. Furthermore, it was observed that the dynamic pressure is affected by the reflections and the emissions from the pressure hull in addition to the incident field from the source point. When the shock wave hits the cross-pressure hull, it is reflected and generates negative pressure. Also, it is interaction with the incident wave decreases the dynamic pressure in the surrounding fluid. This is attributed to the fact that the water cannot withstand tension. That is why the total dynamic pressure acquired negative values as illustrated in the figures. Furthermore, cavitation occurs immediately after the incident shock wave hits the MICEPH. Additionally, local cavitation around the MICEPH was observed once the acoustic pressure declines to the steam pressure of the fluid. Subsequently, when the local cavitation disappears, the load of the fluid on the pressure hull generates vibrations. Additionally, Figure 9 illustrates that the shock wave propagates symmetrically from the stand-off point to the aft and fore,

while it expands in the perpendicular direction. The aforementioned results are in accordance with the results reported in [2,43,59]. Also, Figure 9 presents the volume changes in gas bubbles. If the oscillating gas bubble is close enough to the pressure hull, the differential pressure will be created. When the bubble decreases in volume (due to resistance to water flow close to the hull), that would result in bubble collapsing onto the hull and producing high speed water jet, which in some instances is capable of destroying or holing the pressure hull.

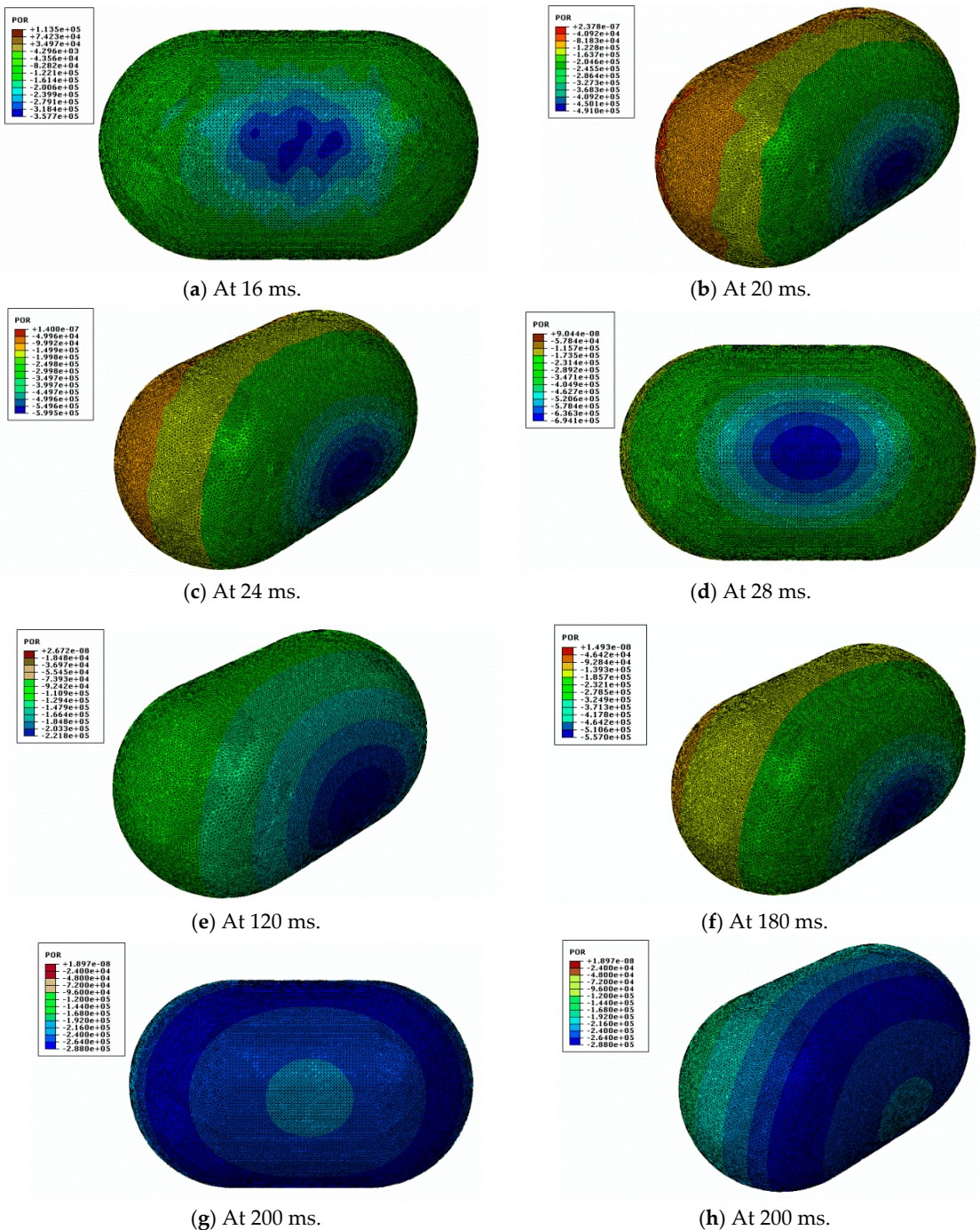

**Figure 8.** Computed POR field distributions at several time instants at the front side of the fluid domain (on the blast side).

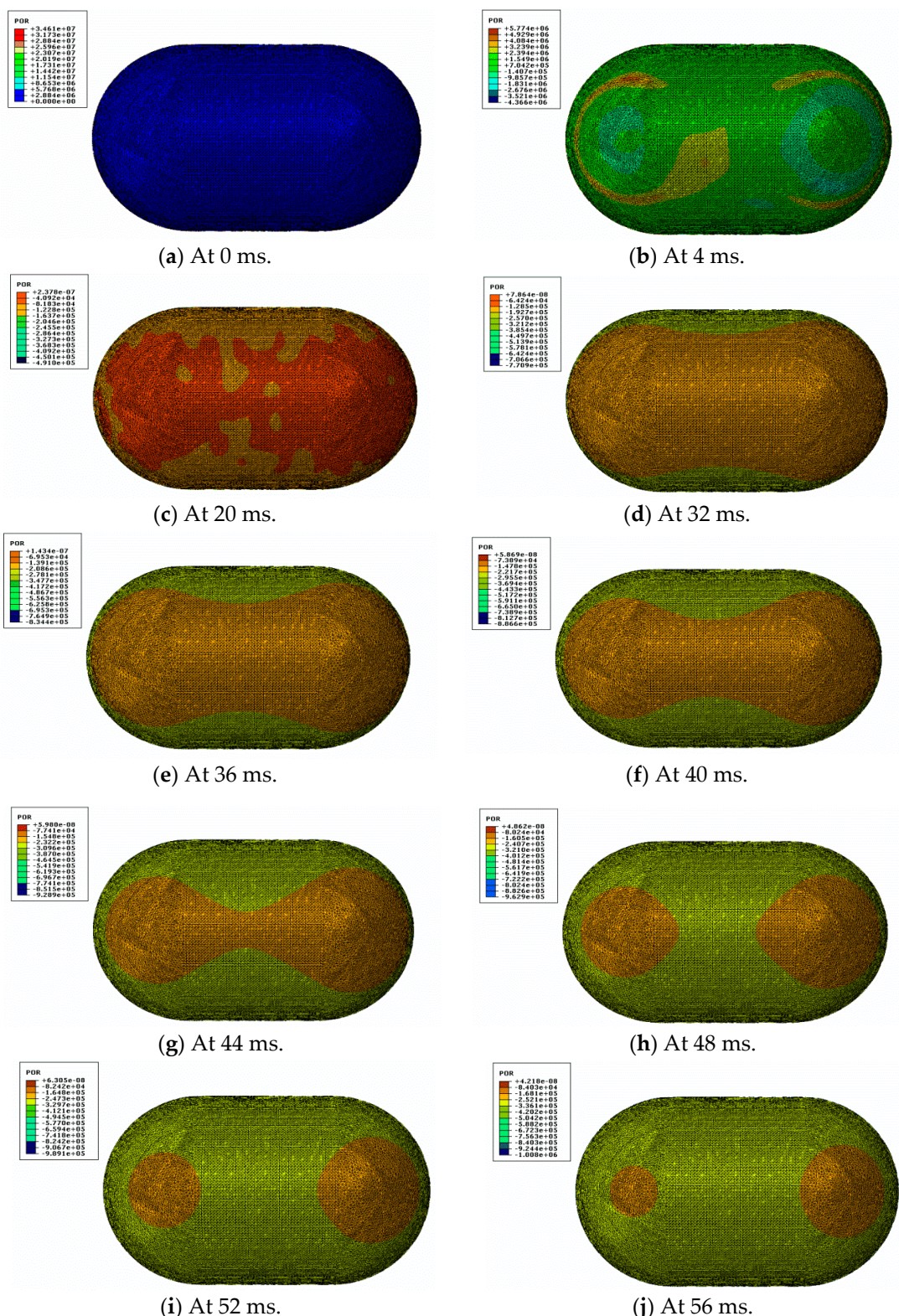

**Figure 9.** Computed POR field at several time instants at the back side of the fluid domain (the side most remote from the charge).

## 6.2. Responses of Submarine Pressure Hull to UNDEX

The optimum MICEPH subjected to non-contact UNDEX will show three major response modes: (i) motion in the axial direction that makes the accordion motion, (ii) motion in a direction at right

angle to the fore-and-aft line of the MICEPH that makes the whipping mode parallel to the direction of the shock wave propagation, and (iii) motion in the vertical direction that makes the breathing motion perpendicular to the shock wave direction of the travel. Several locations were chosen in the model (location *A*, *B*, *C*, *D*, *E* and *F*) to demonstrate the responses in the MICEPH as illustrated in Figure 10.

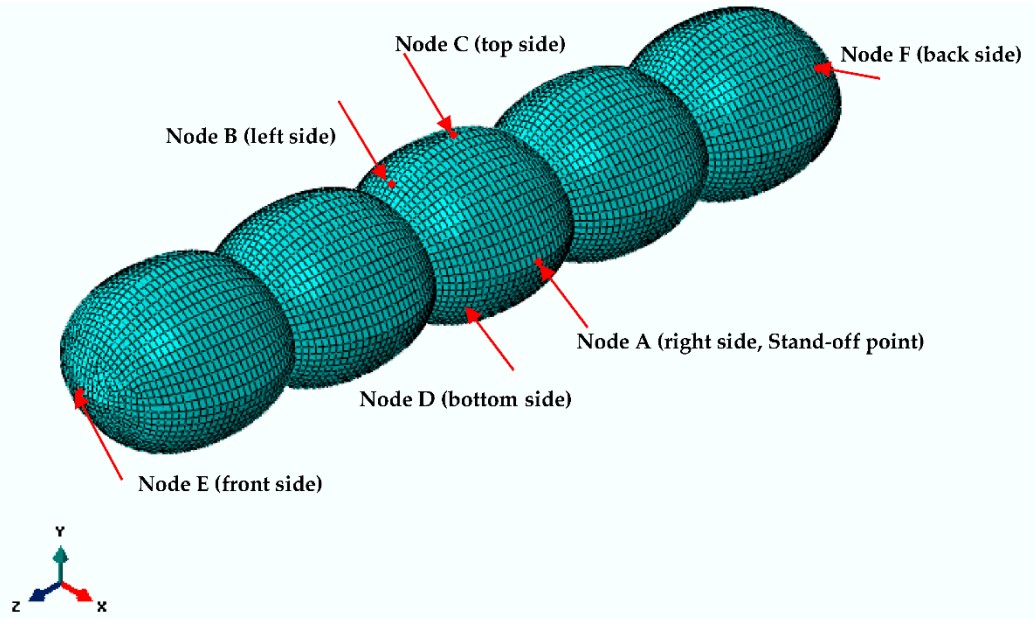

**Figure 10.** Finite element model of Multiple. Intersecting Cross Elliptical Pressure Hull (MICEPH) and the locations of different test points.

### 6.2.1. The Acceleration Response at Different Locations

First, the response of the MICEPH in the axial direction is illustrated in Figure 11. The figure presents the time history response of *Z*-axial acceleration ($A_3$). UNDEX of 20 kg TNT at different offset distances of 5.5 m, 15.5 m and 20.5 m were discussed. At points E and F, located at the center of each end of the MICEPH, it was observed that the acceleration responses ($A_3$) were in the opposite directions and occurs at the same instance. The peak value of the acceleration is $26 \times 10^3$ m/s$^2$ at an offset distance of 5.5 m, and occurs at 4 m/s. This measurement then oscillates and decays after 4 m/s. In addition, while the offset distance increases, the acceleration decreases and acquires $8.3 \times 10^3$ m/s$^2$ and $6.65 \times 10^3$ m/s$^2$ at offset distances of 15.5 m and 20.5 m, respectively. This is attributed to increasing the standoff distance of the explosive charge. Also, the figure demonstrates that the bubble pulses show a minor impact on nodes *E* and *F*.

Similarly, Figure 12 plots the time history curves of *Z*-axial acceleration ($A_3$) for nodes *C* and *D*. The curves illustrate that while the pressure hull stroked by the shock wave, node *C* moves to the left and node *D* moves to the right at the same time instance which causes the accordion motion. This is attributed to the propagated compressive pressure and its subsequent release on the MICEPH in the axial direction. The peak value at node *D* of the measured acceleration is $27.4 \times 10^3$ m/s$^2$ in the negative *Z*-direction, owing to a 20 kg TNT charge and an offset distance of 5.5 m. Likewise, at node *C*, the peak value of acceleration is $23.9 \times 10^3$ m/s$^2$, which occurs at 2 m/s. Also, as the offset distance increases at node *C*, the acceleration decreases and achieves $4.6 \times 10^3$ m/s$^2$ and $2.8 \times 10^3$ m/s$^2$ at offset distances of 15.5 m and 20.5 m, respectively. Similarly, the maximum acceleration at node *D* are $5.1 \times 10^3$ m/s$^2$ and $3.2 \times 10^3$ m/s$^2$ at offset distances of 15.5 m and 20.5 m, respectively. The aforementioned results demonstrate that the peak acceleration at nodes *C* and *D* were achieved before nodes *E* and *F*.

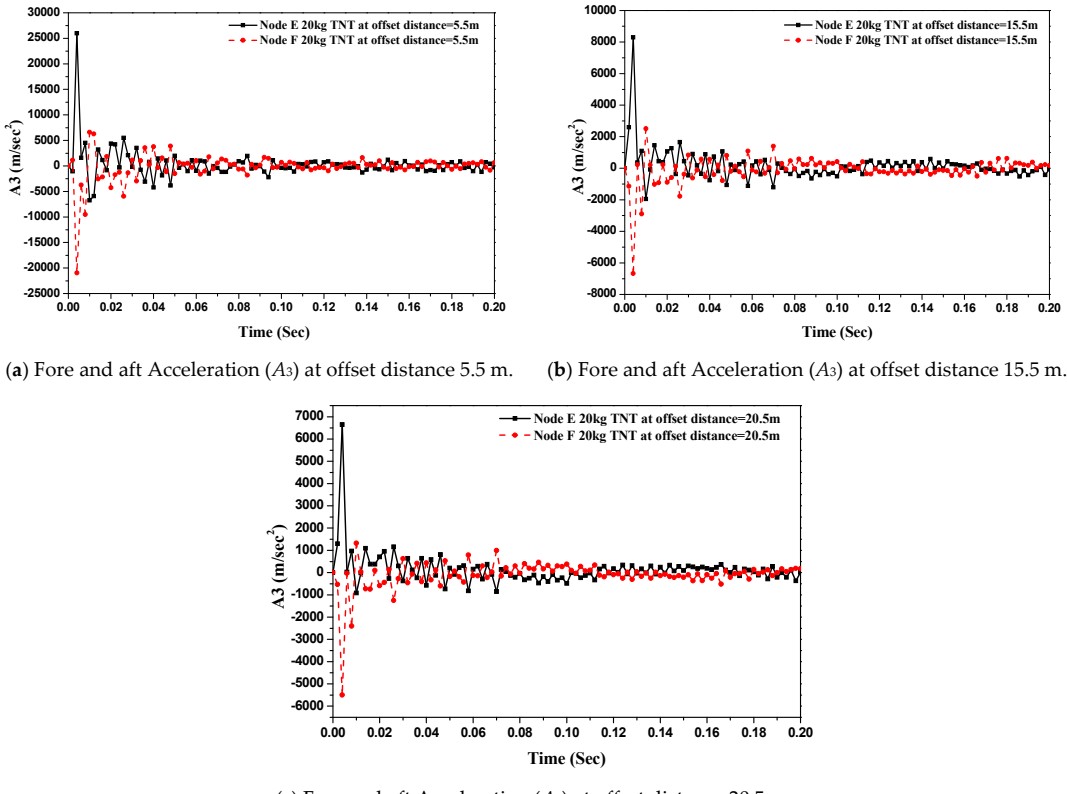

(**a**) Fore and aft Acceleration (*A*3) at offset distance 5.5 m.    (**b**) Fore and aft Acceleration (*A*3) at offset distance 15.5 m.

(**c**) Fore and aft Acceleration (*A*3) at offset distance 20.5 m.

**Figure 11.** The acceleration-time history ($A_3$) at nodes *E* and *F* due to 20 kg TNT on the pressure hull.

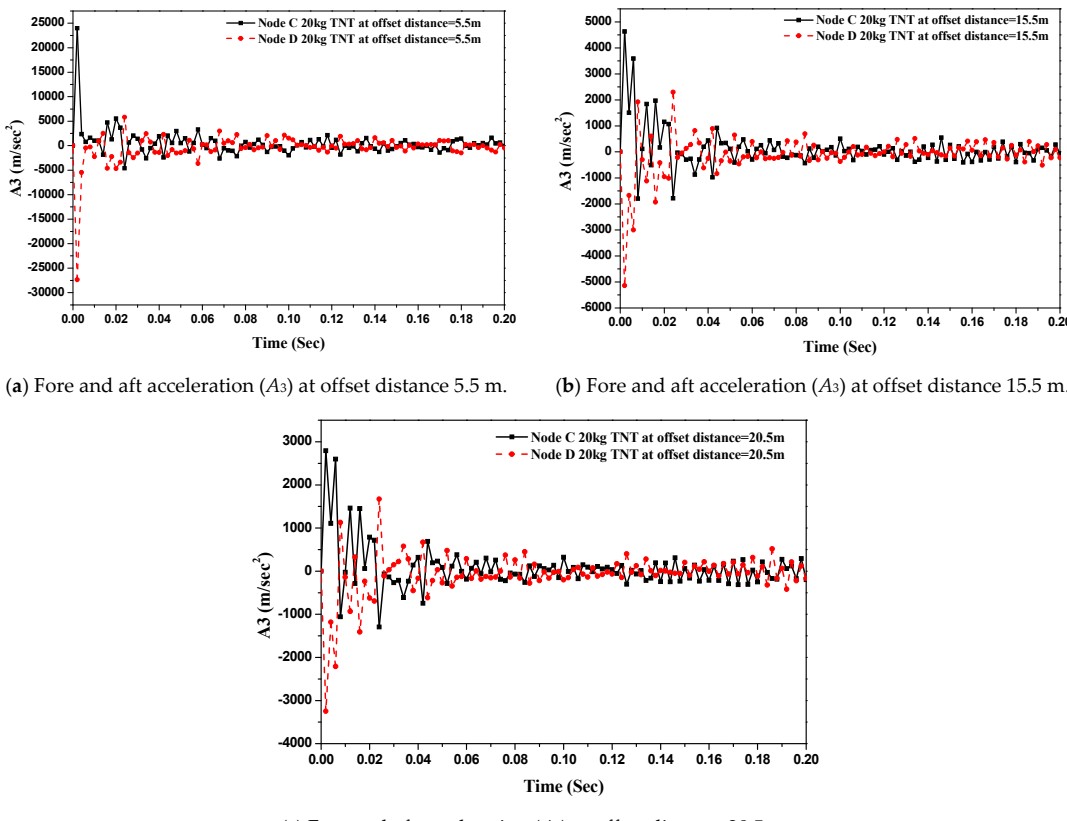

(**a**) Fore and aft acceleration (*A*3) at offset distance 5.5 m.    (**b**) Fore and aft acceleration (*A*3) at offset distance 15.5 m.

(**c**) Fore and aft acceleration (*A*3) at offset distance 20.5 m.

**Figure 12.** The acceleration-time history ($A_3$) at nodes *C* and *D* due to 20 kg TNT on the pressure hull.

On the other hand, Figure 13 illustrates the acceleration response ($A_2$) of the MICEPH in the vertical direction of nodes $E$ and $F$ under UNDEX of 20 kg TNT at offset distances of 5.5 m, 15.5 m and 20.5 m. It is observed that node E moves opposite to node $F$ throughout the transient response of the MICEPH. Also, the figure illustrates the breathing motion caused by the expansion and subsequent contraction of the MICEPH. For instance, at node $E$ and an offset distance of 5.5 m, the peak acceleration ($A_2$) is $9.45 \times 10^3$ m/s$^2$ in the $Y$-direction, while at node $F$, the peak acceleration is $8.96 \times 10^3$ m/s$^2$ in the negative $Y$-direction and occurs at a time instance of 8 m/s. Furthermore, at higher offset distances of 15.5 m and 20.5 m, the peak measured accelerations are $7.16 \times 10^3$ m/s$^2$ and $5.26 \times 10^3$ m/s$^2$ in negative $Y$-direction at node $E$. While at node $F$, the peak accelerations are $6.43 \times 10^3$ m/s$^2$ and $5.15 \times 10^3$ m/s$^2$ in the $Y$-direction and occur at a time instance of 2 m/s.

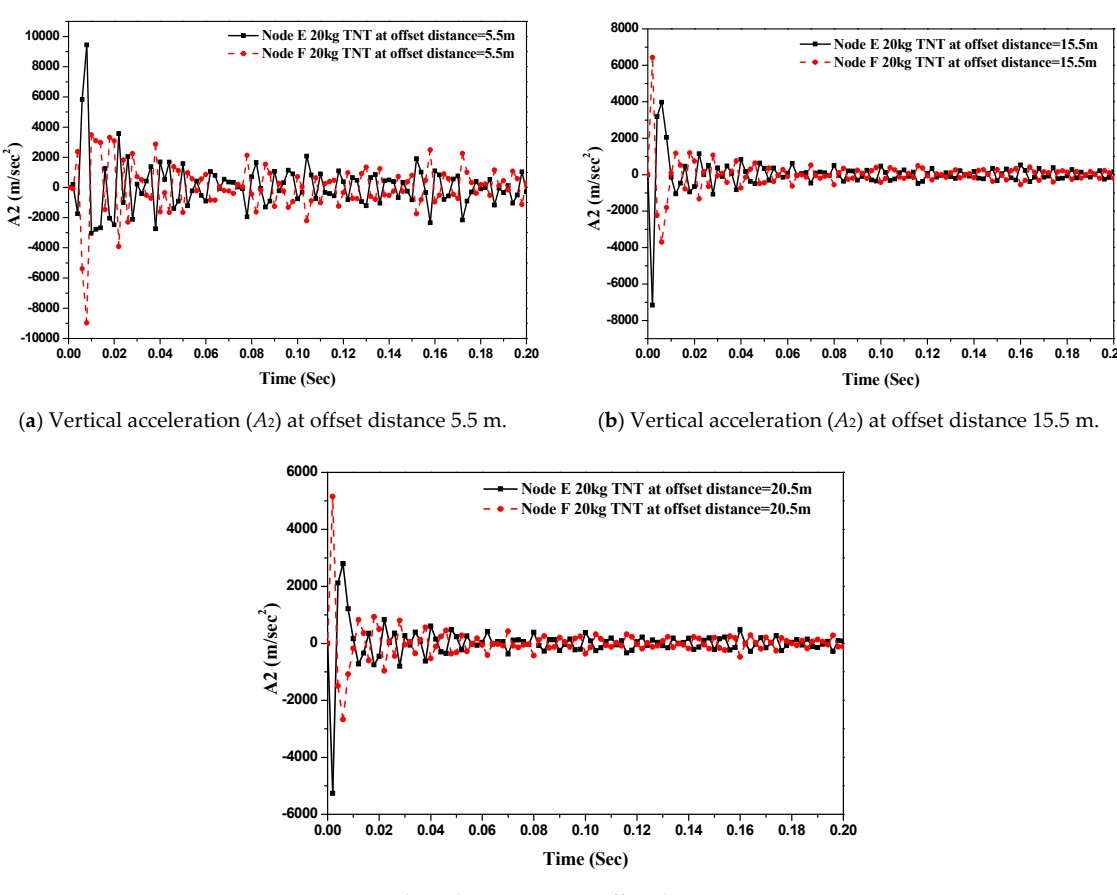

(**a**) Vertical acceleration ($A_2$) at offset distance 5.5 m.

(**b**) Vertical acceleration ($A_2$) at offset distance 15.5 m.

(**c**) Vertical acceleration ($A_2$) at offset distance 20.5 m.

**Figure 13.** The acceleration-time history ($A_2$) at nodes $E$ and $F$ due to 20 kg TNT on the pressure hull.

Furthermore, Figure 14 shows the acceleration time histories ($A_2$) in the vertical direction for nodes $C$ and $D$ which located at top and bottom of the MICEPH under the effect of 20 kg TNT charge located at offset distances of 5.5 m, 15.5 m and 20.5 m. It is revealed that the upper point (node $C$) moves in a direction opposite to that of the lower point (node $D$). The peak accelerations at node $C$, measured downward, are $-40 \times 10^3$ m/s$^2$, $-10.4 \times 10^3$ m/s$^2$, and $-7.37 \times 10^3$ m/s$^2$ at offset distances of 5.5 m, 15.5 m, and 20.5 m, respectively. Likewise, the peak accelerations at node $D$, measured upward, are $49.2 \times 10^3$ m/s$^2$, $12.6 \times 10^3$ m/s$^2$, and $8.25 \times 10^3$ m/s$^2$ at offset distances of 5.5 m, 15.5 m and 20.5 m, respectively. The peak acceleration time histories ($A_2$) at node $D$ are higher than its counterpart at node C, while the frequencies of accordion and breathing motion are nearly the same. These results ensure that the accordion and breathing motion are directly correlated as per [2].

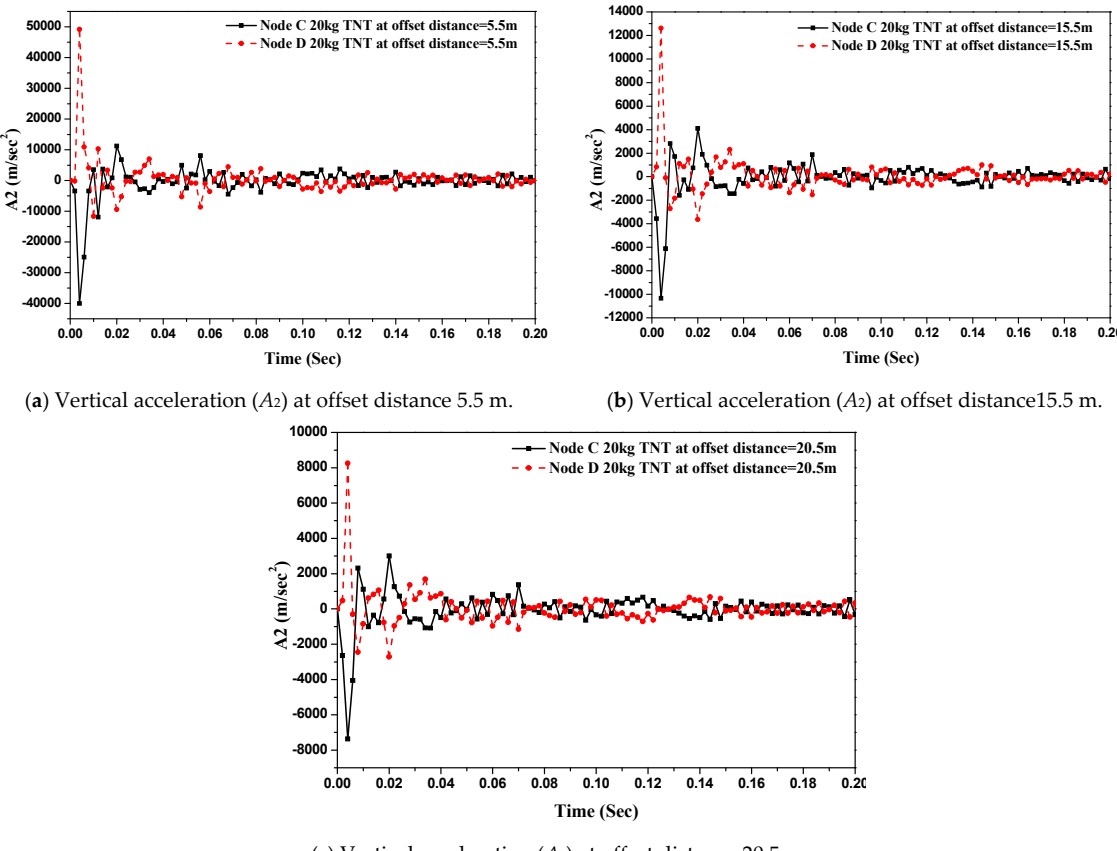

(**a**) Vertical acceleration ($A_2$) at offset distance 5.5 m.

(**b**) Vertical acceleration ($A_2$) at offset distance 15.5 m.

(**c**) Vertical acceleration ($A_2$) at offset distance 20.5 m.

**Figure 14.** The acceleration-time history ($A_2$) at nodes *C* and *D* due to 20 kg TNT on the pressure hull.

Similarly, Figure 15 shows the acceleration time histories ($A_2$) at nodes *A* and *B*. It is observed that, once the shock wave struck the stand-off point at node *A*, the vertical movements of nodes *A* and *B* follow the same direction. Node *A* moves first, followed by node *B*. The peak acceleration at node *A* is $-8 \times 10^3$ m/s$^2$, $-1.8 \times 10^3$ m/s$^2$ and $-1.3 \times 10^3$ m/s$^2$ at offset distance of 5.5 m, 15.5 m and 20.5 m, respectively, and occur downward. Likewise, the peak acceleration at node *B* is $-3.5 \times 10^3$ m/s$^2$, $-1.1 \times 10^3$ m/s$^2$ and $-0.81 \times 10^3$ m/s$^2$ at offset distance of 5.5 m, 15.5 m and 20.5 m, respectively, following the same direction at point *A*. Figure 16 plots the athwart acceleration time histories ($A_1$) at nodes *A* and *B* which is the primary direction of shock wave propagation. It is revealed that the shock wave arrives at node *A* first then hits node *B*. The maximum athwart acceleration ($A_1$) at node *A* occurs in the horizontal direction and its peak is about $52.4 \times 10^3$ m/s$^2$ at an offset distance of 5.5 m. This is attributed to the release of the shock wave, occurrence of local cavitation and effect of bubble pulse. On the other hand, the second peak values, measured at offset distances of 5.5 m, 15.5 m and 20.5 m, occur owing to the uploading of the MICEPH. Similarly, the maximum athwart acceleration ($A_1$) at node *B* also occurs in the horizontal direction, and its peak is about $29.4 \times 10^3$ m/s$^2$, $10.9 \times 10^3$ m/s$^2$ and $7.9 \times 10^3$ m/s$^2$ at offset distance of 5.5 m, 15.5 m and 20.5 m, respectively. From the aforementioned analysis, it is concluded that the local cavitation has a major effect on athwart acceleration at standoff point at node *A*. Likewise, the uploading of the structure and the first bubble pulse also have a major effect on athwart acceleration. Additionally, for nodes *A* and *B*, the transverse responses are very severe at these locations, and the greatest acceleration occurs in the athwart direction, which is the main direction of shock wave with values of $52.3 \times 10^3$ m/s$^2$ and $29.4 \times 10^3$ m/s$^2$ at time intervals of 2 m/s and 4 m/s, respectively, followed by the vertical and longitudinal directions. While, for nodes *C* and *D*, which located at the top and bottom of MICEPH, the peak acceleration occurs in the vertical direction with values of $-40 \times 10^3$ m/s$^2$ and $49.2 \times 10^3$ m/s$^2$ at time interval of 4 m/s, followed by the athwart and longitudinal directions. Furthermore, at nodes *E* and *F*, which located at the center of each

end of the MICEPH (fore and aft direction), the peak acceleration occurs in the longitudinal direction followed by the vertical and athwart directions.

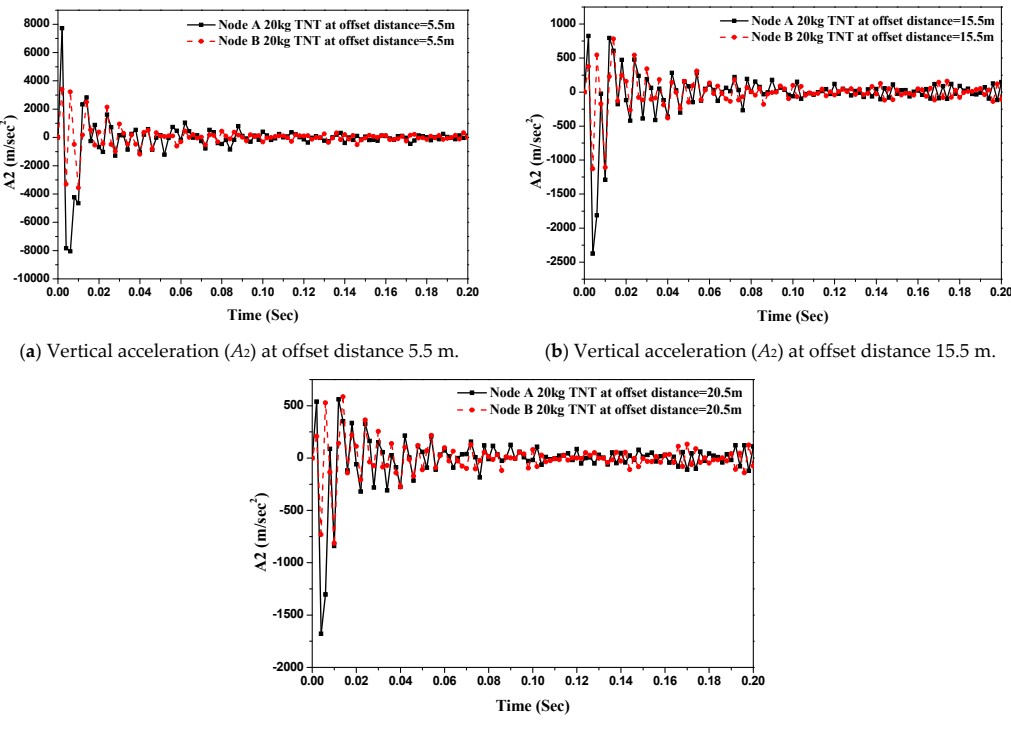

(**a**) Vertical acceleration ($A_2$) at offset distance 5.5 m. (**b**) Vertical acceleration ($A_2$) at offset distance 15.5 m.

(**c**) Vertical acceleration ($A_2$) at offset distance 20.5 m.

**Figure 15.** The acceleration-time history ($A_2$) at nodes *A* and *B* due to 20 kg TNT on the pressure hull.

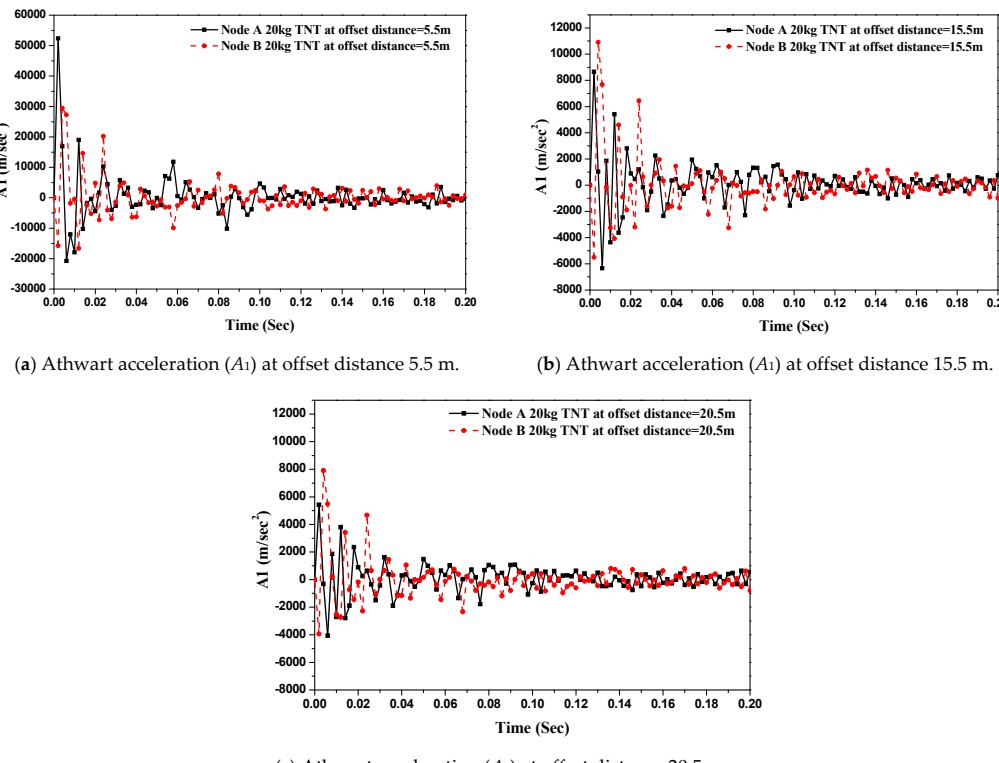

(**a**) Athwart acceleration ($A_1$) at offset distance 5.5 m. (**b**) Athwart acceleration ($A_1$) at offset distance 15.5 m.

(**c**) Athwart acceleration ($A_1$) at offset distance 20.5 m.

**Figure 16.** The acceleration-time history ($A_1$) at nodes *A* and *B* due to 20 kg TNT on the pressure hull.

### 6.2.2. The Displacement Response at Different Locations

Figure 17 illustrates the deformation response in the translation direction ($U_2$) at nodes $A$ and $B$ owing to 20 kg TNT at various offset distances (5.5 m, 15.5 m and 20.5 m). It is observed that the vertical displacement ($U_2$) at nodes $A$ and $B$ move in opposite directions throughout the transient response and follow the same variation. In addition, the figure shows a rigid body translation of the MICEPH. Figure 17a shows the high-frequency response measured at nodes $A$ and $B$, at an offset distance of 5.5 m. It is revealed that at a small offset distance, the structural wavelengths are longer than the acoustic wavelengths. Additionally, the surrounding fluid acts on the structure as a simple damping mechanism and the energy transported away from the structure through the acoustic radiation. Furthermore, Figure 17b illustrates the intermediate-frequency response at nodes $A$ and $B$, at an offset distance of 15.5 m. At an intermediate offset distance, the structural wavelengths are nearly similar to the acoustic wavelengths. In addition, the surrounding fluid added mass and radiation damping on the structure of the hull. Moreover, Figure 17c demonstrates the low-frequency response at nodes A and B, at offset distance of 20.5 m. It is revealed that the structural wavelengths are nearly shorter than the acoustic wavelengths and the surrounding fluid added effective mass to the structure of the hull on the wetted interface. Figure 18 presents the translation ($U_2$) in the vertical direction of two points located at the top (node $C$) and bottom (node $D$) of the amid-ship section at different offset distances (5.5 m, 15.5 m and 20.5 m) due to 20 kg TNT. It is observed that the upper point (node $C$) is moved in a direction opposite to the lower point (node $D$). The figure also illustrates that the displacement at node $C$ is higher than its counterpart at node $D$. In addition, the pressure hull shows an elastic deformations and rigid body motions with a significant heaving and pitching response. These results match those achieved by [60]. Similarly, Figure 19 illustrates the axial displacement response ($U_2$) in the vertical direction of points located at the center of each end of the MICEPH ($E$ and $F$). It is observed that the two nodes move in opposite direction with almost similar frequencies forming the accordion motion. Additionally, the shape of the displacement response ($U_2$) is significantly affected by the offset distance. While the offset distance increases, the frequency decreases. The aforementioned results agree well with that acquired by [40,43].

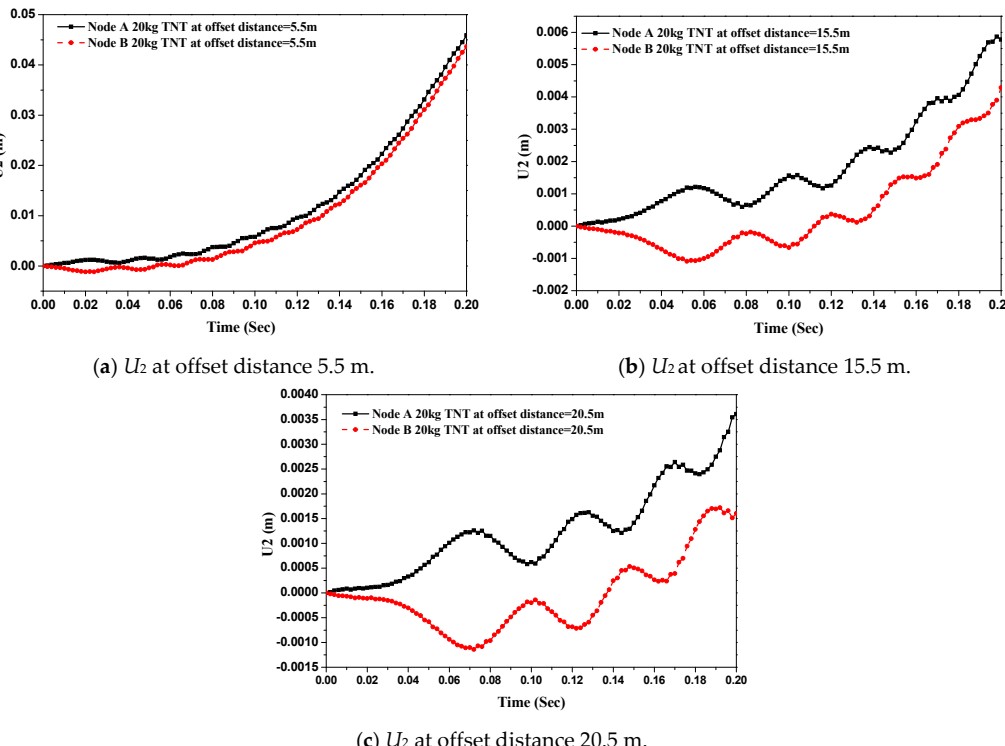

(**a**) $U_2$ at offset distance 5.5 m.

(**b**) $U_2$ at offset distance 15.5 m.

(**c**) $U_2$ at offset distance 20.5 m.

**Figure 17.** Displacement-time history responses ($U_2$) at nodes $A$ and $B$ on the MICEPH.

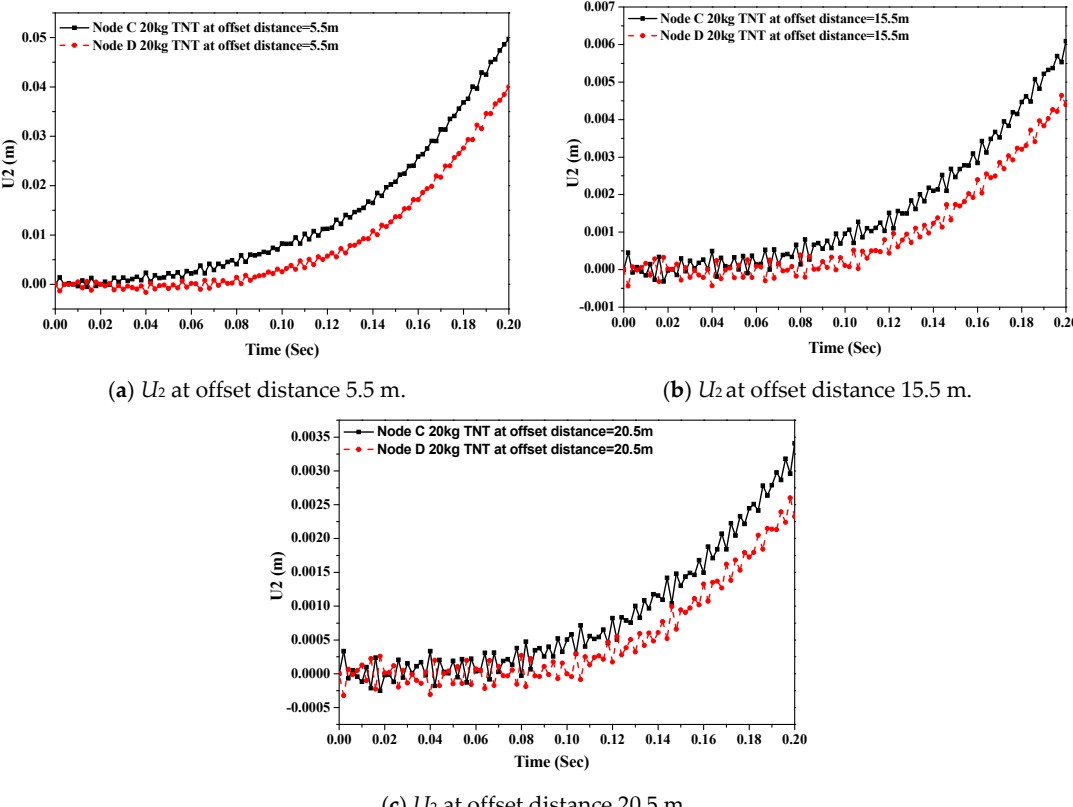

(**a**) $U_2$ at offset distance 5.5 m.

(**b**) $U_2$ at offset distance 15.5 m.

(**c**) $U_2$ at offset distance 20.5 m.

**Figure 18.** Displacement-time history responses ($U_2$) at nodes *C* and *D* on the MICEPH.

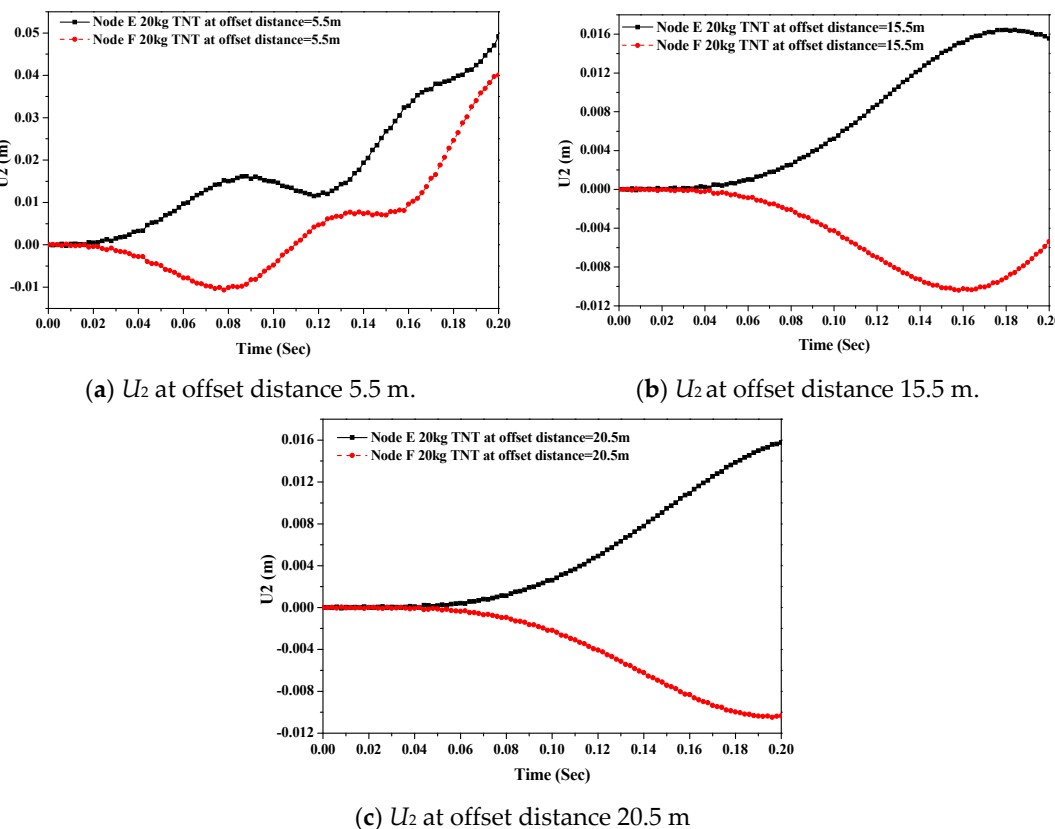

(**a**) $U_2$ at offset distance 5.5 m.

(**b**) $U_2$ at offset distance 15.5 m.

(**c**) $U_2$ at offset distance 20.5 m

**Figure 19.** Displacement-time history responses ($U2$) at nodes *E*, *F* on the MICEPH.

### 6.2.3. Failure Analysis of MICEPH

To evaluate the maximum allowable load that can be sustained before lamina failure, an appropriate failure criterion is needed. Therefore, the failure of the MICEPH is analyzed in this study using different failure criteria (maximum stress and Tsai-Hill failure criteria). The failure was assessed through the calculation of the failure index. If the failure index achieved unity, then the material had failed. Figures 20 and 21 demonstrate the failure index distribution on the structure of the pressure hull due to 20 kg TNT at different offset distances of 5.5 m and 15.5 m at various time instants.

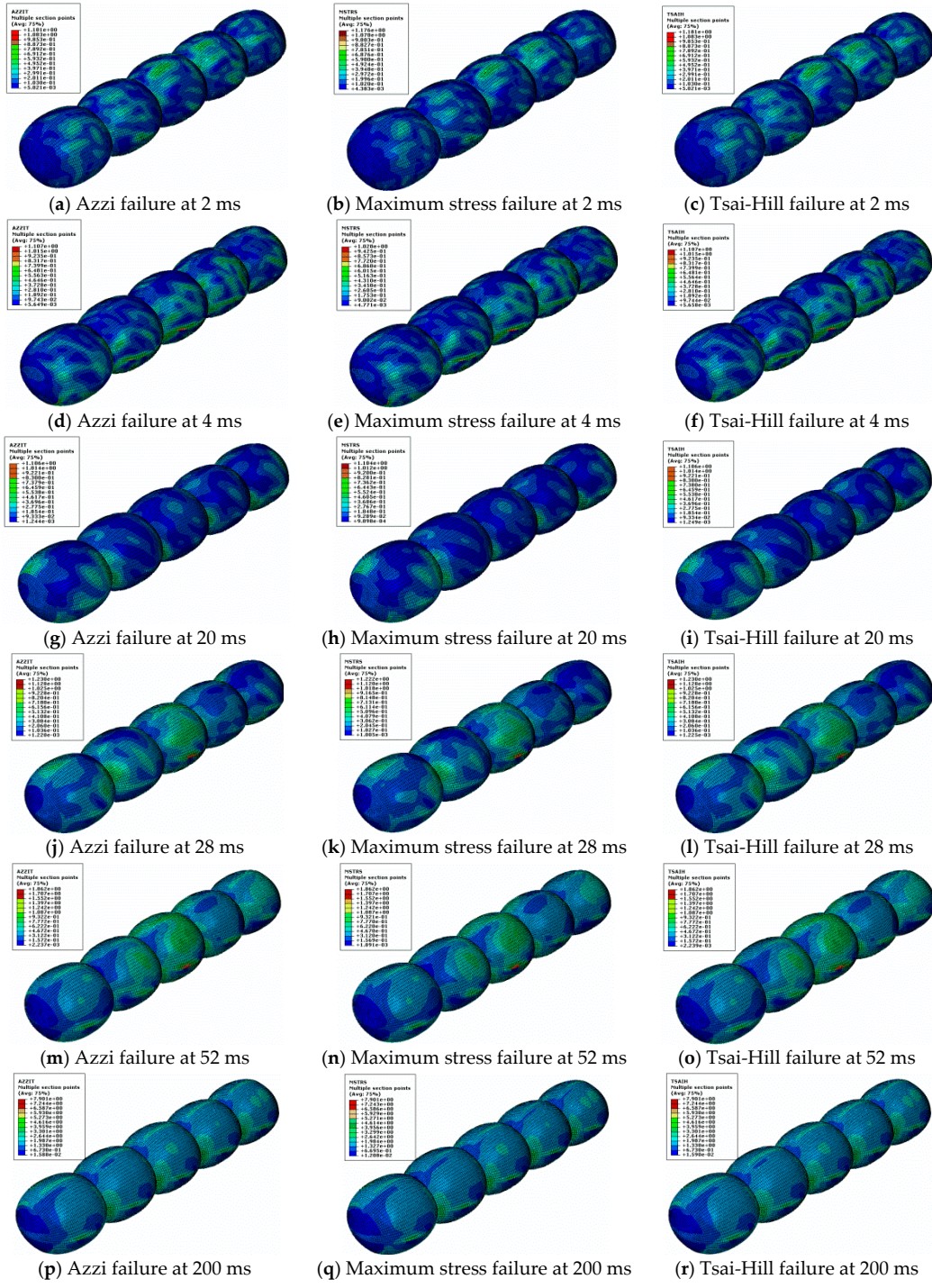

(**a**) Azzi failure at 2 ms    (**b**) Maximum stress failure at 2 ms    (**c**) Tsai-Hill failure at 2 ms

(**d**) Azzi failure at 4 ms    (**e**) Maximum stress failure at 4 ms    (**f**) Tsai-Hill failure at 4 ms

(**g**) Azzi failure at 20 ms    (**h**) Maximum stress failure at 20 ms    (**i**) Tsai-Hill failure at 20 ms

(**j**) Azzi failure at 28 ms    (**k**) Maximum stress failure at 28 ms    (**l**) Tsai-Hill failure at 28 ms

(**m**) Azzi failure at 52 ms    (**n**) Maximum stress failure at 52 ms    (**o**) Tsai-Hill failure at 52 ms

(**p**) Azzi failure at 200 ms    (**q**) Maximum stress failure at 200 ms    (**r**) Tsai-Hill failure at 200 ms

**Figure 20.** Failure criteria distributions on pressure hull due to 20 kg TNT and offset distance 5.5 m.

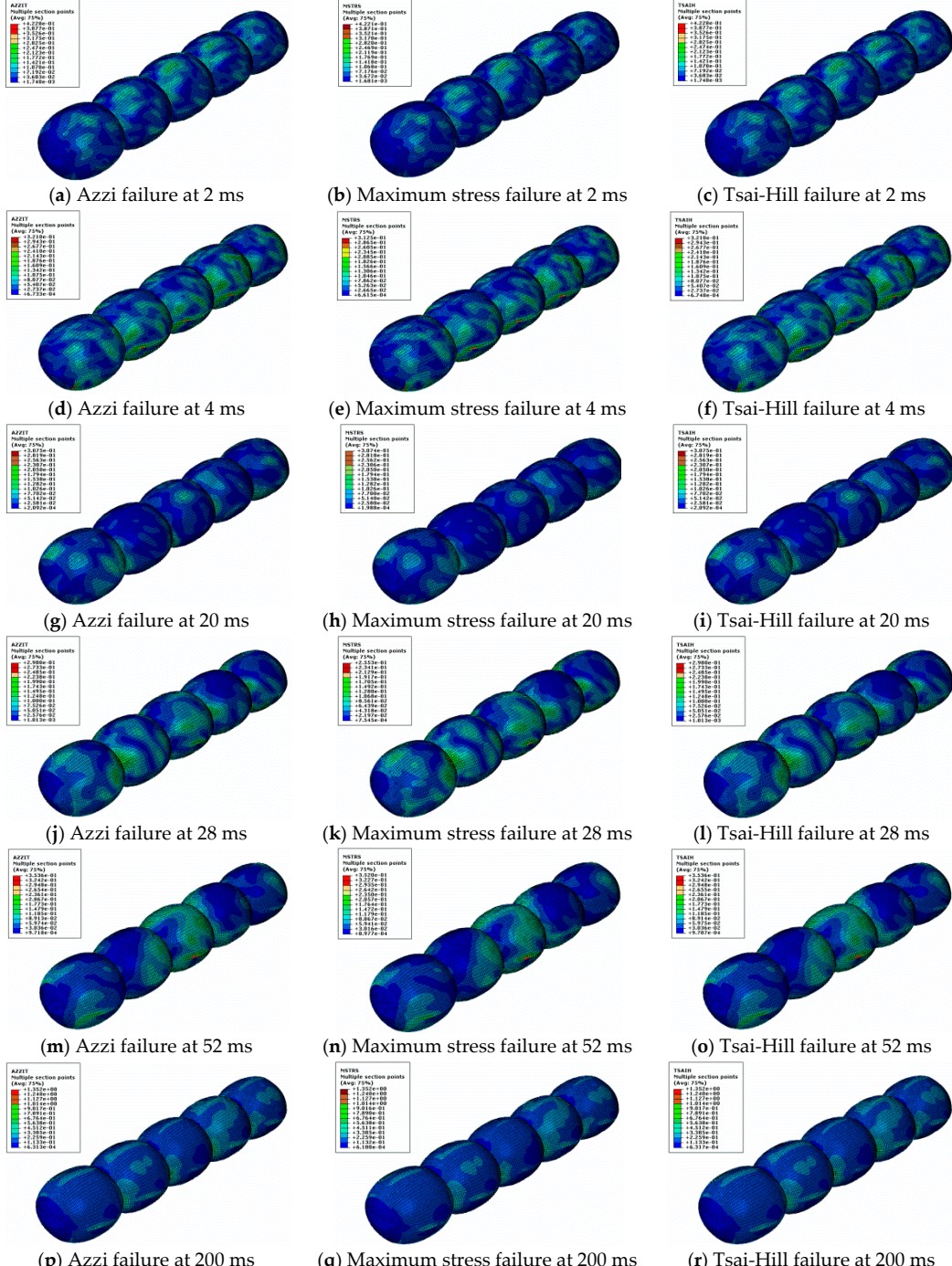

(**a**) Azzi failure at 2 ms    (**b**) Maximum stress failure at 2 ms    (**c**) Tsai-Hill failure at 2 ms

(**d**) Azzi failure at 4 ms    (**e**) Maximum stress failure at 4 ms    (**f**) Tsai-Hill failure at 4 ms

(**g**) Azzi failure at 20 ms    (**h**) Maximum stress failure at 20 ms    (**i**) Tsai-Hill failure at 20 ms

(**j**) Azzi failure at 28 ms    (**k**) Maximum stress failure at 28 ms    (**l**) Tsai-Hill failure at 28 ms

(**m**) Azzi failure at 52 ms    (**n**) Maximum stress failure at 52 ms    (**o**) Tsai-Hill failure at 52 ms

(**p**) Azzi failure at 200 ms    (**q**) Maximum stress failure at 200 ms    (**r**) Tsai-Hill failure at 200 ms

**Figure 21.** Failure criteria distributions on pressure hull due to 20 kg TNT and offset distance 15.5 m.

Generally, it is observed that the failure indices decrease with increasing stand-off distance. Furthermore, the maximum failure indices are measured around the ring and long beams. Moreover, it is revealed that failure is firstly initiated near the stand-off point. Figure 21 also shows the distribution of different failure criteria such as Azzi-Tsai-Hill, maximum stress, and Tsai-Hill criteria on the MICEPH. The results demonstrate that the difference among the different failure criteria is relatively too small. Figure 22 presents the maximum stress and Tsai-Hill failure-time histories for elements *A*, *B*, *C*, *D*, *E* and *F* on the structure of the pressure hull. The results illustrate that the maximum failure index occurs at offset distance of 5.5 m. The failure index owing to the maximum stress and Tsai-Hill criteria showed a similar trend with relatively small differences.

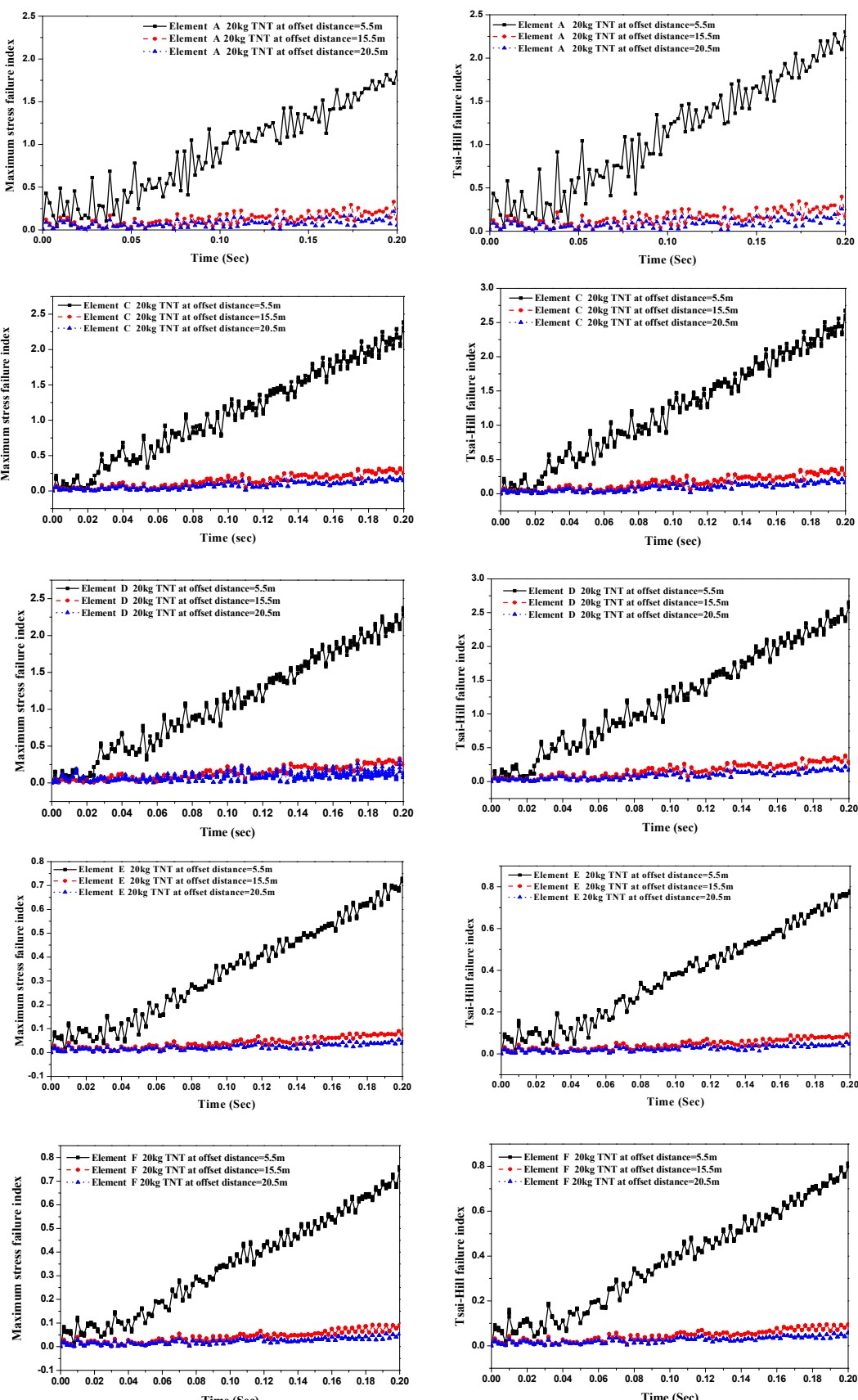

**Figure 22.** Maximum stress and Tsai-Hill failure-time history for elements *A*, *B*, *C*, *D*, *E* and *F* on the pressure hull.

### 7. Conclusions

In this study, the dynamic response of optimized multiple intersecting MICEPH under the effect of non-contact UNDEX was explored. The simulation technique was followed to avoid the expenses and complexity of physical tests. First, the multiple intersecting MICEPH subjected to hydrostatic pressure was optimized. Thereafter, using the optimum design results, the numerical simulation was carried out. Then, the response modes, breathing, accordion and whipping for the MICEPH subjected to non-contact UNDEX were discussed. Furthermore, the effects of applying various failure criterions such as maximum stress, Azzi-Tsai-Hill, and Tsai-Hill criteria for the damage initiation on failure strengths of pressure hull were studied. Based on the analysis and the simulation results, the following conclusions were drawn:

- The explosion weight and standoff distance greatly affects the computed POR field distribution, propagation and its magnitude.
- Based on the study of response modes, for nodes *A* and *B*, the greatest acceleration occurs in the athwart direction, which is the main direction of the shock wave, followed by the vertical and longitudinal directions. Meanwhile, for nodes *C* and *D*, which are located at the top and the bottom of the MICEPH, the greatest acceleration occurs in the vertical direction followed by the athwart and longitudinal directions. Likewise, for nodes *E* and *F*, which are located at the center of each end of the MICEPH (fore and aft) directions, the greatest acceleration occurs in the longitudinal direction, followed by the vertical and athwart directions.
- The cavitation occurs immediately after the incident shock wave hits the MICEPH.
- The local cavitation has a major effect on athwart acceleration. Additionally, the uploading of structure and the first bubble pulse have also, a major effect on athwart acceleration.
- This analysis can predict the failure index of the optimized pressure hull, which can be effectively used to determine the safe explosion weights and standoff distance to avoid the failure.
- The failure index of the pressure hull, caused by the shock wave due to detonation of 20 kg TNT charge at offset distance 5.5 m away from the side of the amidships of the hull is higher than one. Therefore, in this case, the pressure hull will collapse.
- On the other hand, at offset distance 15.5 m and 20.5 m, the failure index of the pressure hull is less than one, and the pressure hull will avoid the failure. Consequently, the standoff distance has a great effect on the failure index.
- The test results showed that applying different failure criterion such as maximum stress, Azzi-Tsai-Hill, and Tsai-Hill criteria for the damage initiation have a slight to no difference among them on the failure strengths of pressure hull.
- The simulation technique utilized in this study and its results can serve as a valuable reference for designers to enhance the resistance of underwater vehicles against underwater explosion.

**Author Contributions:** E.F. and M.H.; methodology, E.F. and M.H.; software, E.F. and M.H.; validation, E.F.; M.E.; M.A.; formal analysis, E.F. and M.H. investigation, H.H. and D.W., M.E., M.A.; resources, E.F. and M.H.; data curation, E.F.; writing—original draft preparation, all, writing—review and editing, E.F.; M.E.; M.A.; E.F.; visualization, E.F. and D.W; supervision, E.F, H.H. and D.W., project administration, D.W.; funding acquisition.

**Funding:** The authors are grateful for the support of this research by the 13th Five Years Key Programs for Science and Technology Development of China (Grant No. 2016YFD0701300), the Chinese Natural Science Foundation (Grant No. 51405076), and Heilongjiang Province Applied Technology Research and Development Program Major Project of China (Grant No. GA16B301). Also, the authors are grateful to Military Technical College (Cairo, Egypt), Taif University (Taif, KSA) and Mansoura University (Mansoura, Egypt) for providing all the required facilities to carry out the present research.

**Acknowledgments:** The authors are grateful to Military Technical College (Cairo, Egypt), Taif University (Taif, KSA) and Mansoura University (Mansoura, Egypt) for providing all the required facilities to carry out the present research.

**Conflicts of Interest:** The authors declare no conflict of interest.

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
