# Peer review of "Numerical Analysis and Dynamic Response of Optimized Composite Cross Elliptical Pressure Hull Subject to Non-Contact Underwater Blast Loading"

_applsci, doi:10.3390/app9173489_

Round 1

Reviewer 1 Report

The authors have addressed numerical analysis and dynamic response of cross elliptical pressure hull under non contact under water blast loading. The abaqus/explicit code was used with solid-fluid interaction. The finite element analysis is performed well and the physics behind the shock waves is explained well with different parameters. The conclusions are supported by the data. However, minor revision is required in terms of spell checks and grammatical errors. The main text in introduction section is missing figure numbers. The POR fields in figures 3a-3d should have same orientations and length scales. Figure 3a shows many different geometric shapes/orientations of color legends. Please be consistent of only one orientation for all figures.

Reviewer 2 Report

Please see in the attached file!
